# Use of IDeS Method to Design an Innovative HYICE Sportscar

Giulio Galiè *, Michele Cappelli, Pietro Maffei, Matteo Robusti, Igor Vasileski and Leonardo Frizziero

DIN—Department of Industrial Engineering, Alma Mater Studiorum—Università di Bologna, 40136 Bologna, Italy
* Correspondence: giulio.galie2@unibo.it

**Abstract:** In the contemporary automobile scene, environmental effect abatement is being increasingly sought; this demands a full rethinking of the entire system and entails more than just the reduction in exhaust pollutant emissions. Currently, the most popular approach is the electrification of automobiles, which significantly reduces pollution in major urban areas while simultaneously posing a new set of problems. The two types of zero-emission vehicles that are now being developed the most are hydrogen fuel cells and battery electric cars, but another option is the Hydrogen Internal Combustion Engine (HYICE) engine, which is highly advantageous in terms of pollutants, aside from Nitrogen Oxides (NOx), which can be considerably decreased. The purpose of this study is to develop a novel vehicle design that transports this type of technology into a sporting context while striving for considerable environmental benefits and integrating them into a society where the love of automobiles still has a strong following. The cutting-edge Industrial Design Structure (IDeS) methodology is used in this work, and a sample structure was created to demonstrate how the problems and technical limitations represented can be solved. The steps of the methodology are followed to shape the final product, with careful consideration given to the design of the styling component through the use of the Stylistic Design Engineering (SDE) method. With the ultimate goal of achieving sustainable driving pleasure, the study looks into whether recyclable materials can be used for the body and whether extremely light materials can be used for the chassis.

**Keywords:** car design; future mobility; HYICE engine; IDeS methodology; recyclable materials





## 1. Introduction

This project was started with the intention of developing a different sports automobile option for the future. Looking forward to the future, we can imagine that pollution restrictions will be increasingly stringent, thus leading to the abandonment of the use of internal combustion engines; with the use of fossil fuels, we will never be able to meet the demand for zero emissions, despite the use of innovative systems that can break them down.

This study therefore aims to analyze a market segment, that of sports vehicles, which could be affected by this dramatic change as only battery electric vehicles and fuel cell electric vehicles, which use chemical energy from hydrogen to convert it into electricity, are currently available as alternatives to internal combustion engines. The sports car segment has a passionate customer base, and in the future, they may have no choice but to drive an electric car. This could create a market with high demand and little or no supply.

Innovation in this segment could arise from a different use of hydrogen, no longer as a chemical energy resource, but as a substitute fuel for common fossil fuels within an internal combustion engine [1,2]. With a similar power delivery and the option to continue using the manual gearbox, it would be possible to create a car with the driving involvement and feelings of a modern sports vehicle. This kind of hydrogen utilization has previously been demonstrated to be feasible in several studies carried out by well-known automakers such as Toyota and BMW.

Due to the tiny size of the hydrogen molecule and the energy-intensive nature of the synthesis, which is presently not practical from an economic standpoint, storage and production are the two key issues facing hydrogen at this time [3]. The distribution of hydrogen is another important issue; however, tests such as those in Italy by Snam have already been conducted for the transportation of hydrogen using the infrastructure utilized for natural gas, which is already in place.

There is a recognized international standard (ASME B31.12) which defines the compatibility of new and existing steel piping for the transport of hydrogen. Currently, the tests carried out guarantee an injection of 10% of hydrogen into a mix with natural gas, and 70% of Snam's methane pipelines are compatible with this type of use.

To date, 23 European gas transmission operators have joined the European Hydrogen Backbone (Figure 1), a new transport network that will extend to 39,700 km in 2040, connecting 21 European countries, with a plan that bases two-thirds of the future network on gas pipelines that already exist [4,5].

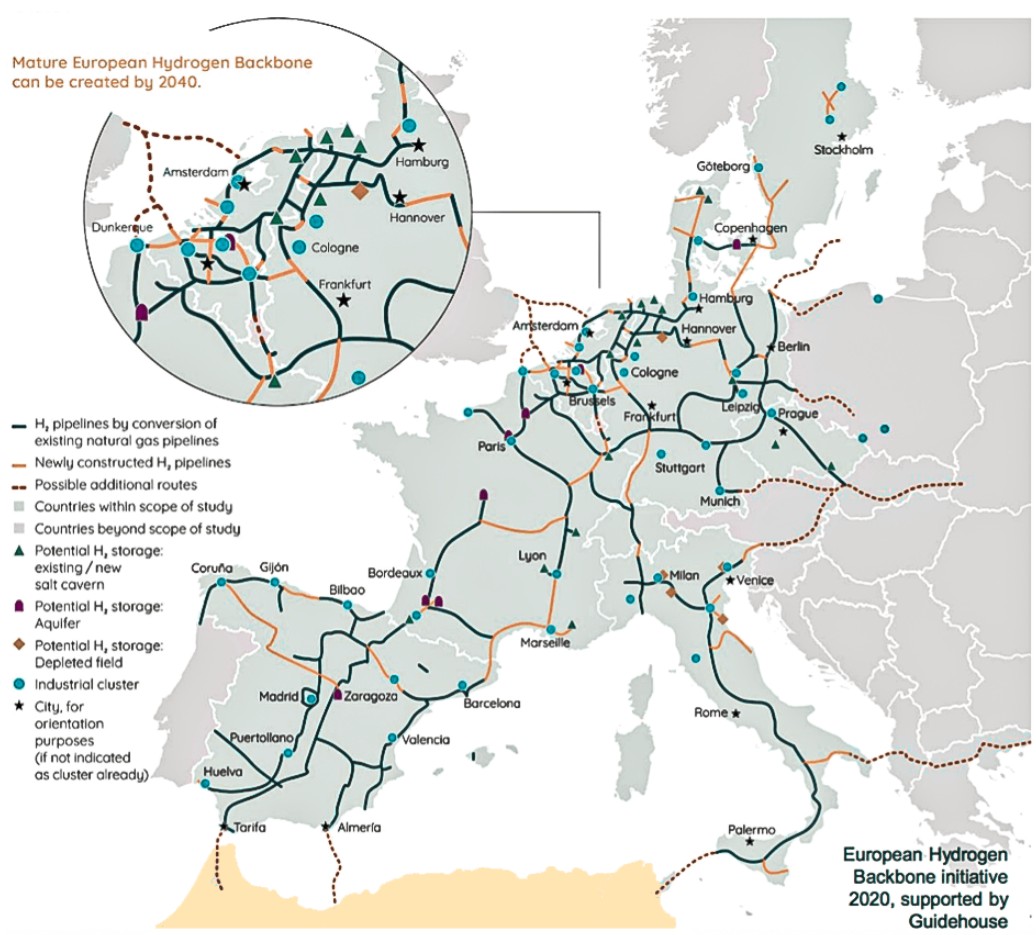

**Figure 1.** Hydrogen infrastructure network.

As a result, the HYICE (hydrogen internal combustion engine) technology would be ideal for usage in sports cars, as opposed to city cars, SUVs, and other typical vehicles, which might make better use of the other technologies to increase efficiency because they do not need to be exciting to drive. The most prevalent gas in the cosmos is hydrogen; however, the crust of the earth does not contain pure hydrogen. It is difficult to obtain hydrogen in its pure state [6]. The two most popular processes for creating hydrogen are electrolysis of water and steam reforming, a high-temperature process in which steam combines with fuels that contain hydrocarbons to create hydrogen. Water is divided into hydrogen and oxygen during the electrolysis process (Figure 2) by running an electric current across it [7].

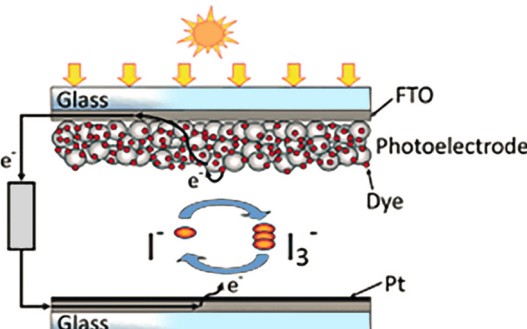

**Figure 2.** Electrolysis process.

Other hydrogen generation techniques have been pursued in recent years in an effort to make this process more environmentally friendly, economically viable, and energy efficient. Currently, electrolysis using energy generated from renewable sources is the best option [8]. The European community is investing heavily to increase hydrogen production in the coming years, and we are confident that with the right economic and scientific interest these problems can be solved.

There have only been limited research efforts (in terms of the money and teams engaged) and demonstration activities after the huge programs of the 2000s, largely by BMW, but with several other automakers involved. Several of them were made before the 2014 review, but they were not taken into consideration. Some additional studies that supplement the review are suggested below [9].

- CI diffusion: A hydrogen CI (compression ignition) ICE (internal combustion engine) engine may perform similarly to a diesel CI ICE, but this is at the expense of further research and development because the hybrid design is still a long way from being ready for mass production. With just few modifications, Westport could convert HPDI (high-pressure direct injection) technology to use hydrogen instead of natural gas. Vehicles and fueling technology using natural gas are currently available. It is possible to convert natural gas cars into cars that run on pure hydrogen or a combination of compressed natural gas and hydrogen. While pure hydrogen DI (direct injection) and hydrogen/CNG (compressed natural gas) mixes were studied in the past by Westport, the findings were largely unreported. The fuel storage, fueling practices, station regulations, rules, and standards are all the same for hydrogen and natural gas [10].

- PI premixed: There has been a report on a hydrogen-powered naturally aspirated PFI (port fuel injected) engine. Turbocharging, numerous fuel injectors per port, and charge dilution control were among the engine changes. Jet Ignition (JI) and high charge dilution through dual independent variable cam timing with no EGR (external exhaust gas recirculation) provide ultra-lean burn and throttle-less control. In experiments, low NO emissions and efficiency close to 38% were observed [11]. Another study considers a cryogenic PFI JI turbocharged engine. The maximum speed range is 7500 rpm. The impacts of air displacement and other drawbacks of PFI with hydrogen are restricted by cryogenic injection. As with a normally aspirated gasoline engine, power densities exist. Running at an ultra-lean $\lambda = 2.32$, the engine generates 80 kW per liter of power and 150 Nm per liter of torque, with acceptable levels of knock index and BSNOx. The efficiency rates are expected to be above 40% and over 35% for the majority of the load-speed map, ranging between $\lambda = 2.32$–3.57. Energy efficiency decreases as the fuel-to-air equivalency ratio increases; however, even the level of $\lambda = 5.56$, in low speed and load circumstances, which are crucial for urban driving, allows for efficiencies of about 30% [12]. The hybrid PI–CI systems produced peak efficiencies at around 42% that were similar to the best diesel ICEs and used surface ignition for diesel combustion. By the time the project was finished, the technology was still in its infancy. A dual fuel diesel–hydrogen engine strategy makes it much

simpler to operate CI with hydrogen. As the Westport HPDI solution for diesel and liquefied natural gas (LNG) has been thoroughly tested over many years, this solution does not require any additional research or development [13].

- CI/PI hybrid: By simply adjusting the quantity of hydrogen injected both before and after the diesel injection ignition or spark-initiated JI, CI dual fuel diesel injection ignition ICE may be operated in a variety of combustion modes. It goes without saying that there are also more or less "controlled" homogeneous charge compression ignition (HCCI) modes of combustion, with diesel injection or spark-initiated JI (jet ignition) occurring before the expected start of the HCCI autoignition, to provide stability to the otherwise unstable HCCI system [14]. Using a direct hydrogen fuel injector and a JI pre-chamber, a diesel truck engine that had been adapted to run on hydrogen was given four different modes of injection and combustion for consideration [15].

JI may take place before the main chamber fuel is injected and mixed with air; the engine then operates using diffusion combustion, which is similar to that of a diesel engine.

When fuel is fed into the main chamber and mixed with air, the engine may run on premixed combustion, which is similar to how gasoline burns.

After air and fuel are combined in a portion of the main chamber and the engine operates with premixed, diffusion, and mixed diesel/gasoline combustion, JI may occur.

Lastly, even if JI does not happen, the engine may still run HCCI combustion; however, if JI does happen suddenly, before the anticipated HCCI start, the engine runs "managed" HCCI combustion.

The design phase was founded on the fundamental concepts of industrial product design, relying on market research to create an attractive and competitive product. To perform an in-depth analysis and innovative product development, we applied the industrial design structure (IDeS) methodology. By exploiting methodologies such as quality function deployment (QFD) and stylistic design engineering (SDE), we were able to obtain an excellent starting point for the development of the actual product. Subsequently, it was possible to exploit the innovative additive manufacturing (AM) technology to obtain rapid prototyping. This makes IDeS an easily improvable and extremely flexible process.

This project also employed the SDE technique to more effectively achieve the objective of innovation in the design industry. It is a design strategy that has been employed effectively in the industrial sector and the auto industry for a long time. The major goal of this study is to provide, using IDeS, which combines QFD and SDE, a technologically new and alternative solution, such as that of HYICE, for a market that is now stable and expanding but is growing quickly.

What greatly differentiates this work from the already existing work is the desire to keep among the design choice factors not only the technical and economic aspects, which certainly lead to the definition of a concept that is as functional as possible, but also the social aspects related to the user experience with endothermic engines; these are still preferred by a substantial portion of consumers, due to the many advantages still offered and the overall experience of use, an aspect amplified in the sports car segment considered for this project.

## 2. Materials and Methods

### 2.1. Environmental Analysis, Study of the Market Segment

Over the decades, customers have sought not only elegance, lightness, harmonious style, dimensions, and modernization from automobiles but also inspiration, as we have seen in the most recent trending markets, and we can imagine that they will do so in the future as well. Moreover, we cannot limit our discussion to aesthetics. Consumers seek things with force, aggression, and responsiveness because they require performance as well. It is clear that customers desire something that pushes the boundaries since, in our opinion, people nowadays are exposed to a variety of stimuli and find it hard to enjoy themselves in the autonomous world [16]. On the environmental front, there is a strong desire to reduce pollution and, as a result, to search for alternatives to the traditional Internal

Combustion Engine (ICE). Customers are increasingly sensitive to issues such as hydrogen, e-fuels, electricity, and anything that can overcome the pollution problem through a more sustainable environmental impact [17]. It is easy to notice the ever-increasing investments into Electric Vehicle (EV) technology to try to overcome those environmental issues [18], but we also noticed how difficult it is to eradicate the emotional aspects of driving an Internal Combustion Engine (ICE) vehicle in people who see cars not only as a mere means of mobility but also as something to enjoy in the form of pure driving pleasure, as a famous car commercial used to say [19]. That is the reason behind the choice to develop a concept vehicle powered by an HYICE engine, a technology in which there is still much to invest in and experiment with, but one that could potentially prove to be a viable alternative on the sustainability front to EV technology, given its known pollution problems stemming from the technology inherent in batteries [20–23]; at the same time, the aim is to appease that segment of the population mentioned earlier. To better approach this segment of the population, and to begin the proposal of this technology, we thought of targeting a niche market [24], that of small sports cars, in order to better meet the tastes of these enthusiasts and to receive more direct feedback on which to develop the concept; we can then proceed to create more articulated proposals in the future on different segments that can go on to make the most of the strengths of this technology.

The goal for this design is a vehicle which is compact, nimble, and attractive due to its sporty performance and its character derived from the propulsion system. As we can see from the graphs (Figures 3 and 4), the numbers in this sector remain constant, with the gradual entry of electric vehicles into this market and the presence, in increasing numbers, of more and more automakers. However, while electric propulsion on the one hand presents very high performance, especially from the point of view of vehicle acceleration, thanks to the very high torque present from the lowest revs, there are also numerous downsides from the point of view of the overall user experience, such as the very high weight, even though it is centralized and distributed at the bottom, the completely absent sound, which severely limits the involvement of the rider during sports driving, and the now well-known problem of range and range anxiety increased by the length of charging times and the dependence on the widespread presence of high-power charging stations in the territory [25].

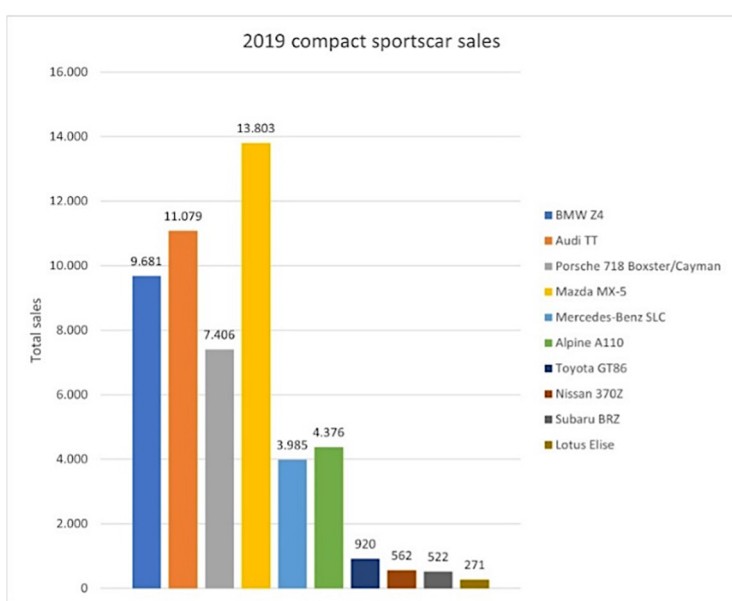

**Figure 3.** Sales in 2019 (source: https://carsalesbase.com/european-sales-2019-exotic-sports-cars/ accessed on 16 Febraury 2022).

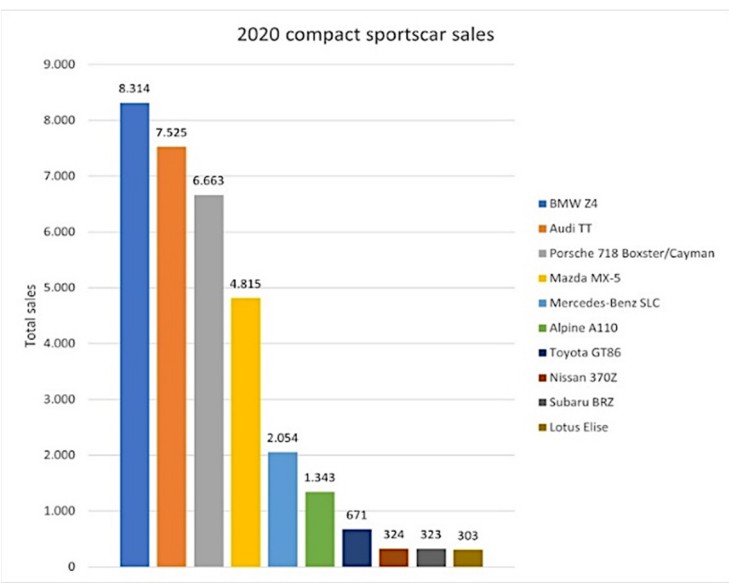

**Figure 4.** Sales in 2020 (source: https://carsalesbase.com/european-sales-2020-exotic-sports-cars/ accessed on 12 March 2022).

Currently, this customer segment might still find a solution in conventional sports cars, but in the future, given the new restrictions on the use of gasoline engines (individual country taxation and EU regulations on $CO_2$ emissions and pollutants tend to become more and more restrictive over time [26]), they might have to settle for a "silent" electric vehicle. This target market could be drawn to something wholly novel, such as a HYICE car. As we have already described, hydrogen engines generate electricity by burning hydrogen using a fuel supply and injection systems modified from those used in petrol engines [27], and in this way, the dispute about driver engagement can be settled with this technology since a HYICE can provide the same convenience and involvement in driving as a regular ICE [28]. Combustion in a hydrogen engine takes place at a faster rate than in a petrol engine, which results in good responsiveness and a potentially better environmental behavior at the same time, as we discussed before. Through sound and vibration, hydrogen engines also have the potential to make driving more enjoyable.

It is clear that combustion engine vehicles dominate the top models in the compact and fun sports car class (Table 1). A wide variety of performance and, consequently, sales prices are available in this area. It is reasonable to think that the lack of a creative model on the market and the range of price options available in this market may favor the design of this new automobile.

**Table 1.** Comparison of the best-selling models of 2020.

|  | Length (mm) | Width (mm) | Height (mm) | Wheelbase (mm) | Engine (cm³) | Powernet (kW) | Torque (Nm) | Curb Weight (kg) | Price (EUR) |
|---|---|---|---|---|---|---|---|---|---|
| BMW Z4 30i | 4324 | 1864 | 1304 | 2470 | 1998 | 190 | 400 | 1415 | 53,000 |
| Audi TT 40 TFSI | 4191 | 1832 | 1355 | 2505 | 1984 | 145 | 320 | 1360 | 44,950 |
| Mazda MX-5 RF | 3915 | 1735 | 1236 | 2310 | 1998 | 135 | 205 | 1072 | 34,500 |
| Porsche Cayman S | 4397 | 1801 | 1284 | 2475 | 2497 | 257.5 | 420 | 1385 | 75,506 |

### 2.2. Benchmarking Analysis

2.2.1. Independence Matrix and Importance Matrix

As we have seen from the environmental analysis, the automotive industry is moving in the direction of reducing pollution and is looking for something that offers alternatives to the classic internal combustion engine, and as we believe that the future is in hydrogen and electricity, we are developing a sports car powered by a hydrogen internal combustion engine. At the moment, the market only offers petrol combustion engines; so, this gives us a new market opportunity. Moreover, this market segmentation is heterogeneous and offers a wide range of performances and values, giving us the opportunity to launch something innovative at a competitive price. To make the most of the analyses necessary for optimal project goal achievement, the IDeS method was used to coordinate all the initial decision-making stages of the project [29]. After getting the answers to the six questions, the QFD approach states that it is time to decide which product would be the best to develop. It is crucial to understand how the answers to the six questions connect to one another [30,31]. The so-called dependence matrix and the relative significance matrix are required for this [32]. As we already know the answers to the six questions, we can write them down in a synthetic form that encapsulates what the customers want. We can then write the synthetic elements in the columns and rows of the dependency matrix and finish it off with the parameters that are defined by responding to the question: "How is an element in a row impacted by an element in a column?".

The parameters are numerical values that allow for the quantification of the dependence between each element in the column and the element in the row. (An empty field denotes a null value.)

- A value of 1 for a weak relationship;
- A value of 3 for a medium relationship;
- A value of 9 for strong dependency.

As a last step, the values of each row and column are added to analyze which parameters are being influenced (Figure 5).

| | Price | Performance | Comfort | Design | Innovation | Safety | Smart | Driving Involvment | Personalization | Total |
|---|---|---|---|---|---|---|---|---|---|---|
| Price | | 1 | 3 | | 3 | | | | 3 | 10 |
| Performance | 1 | | 1 | 9 | 9 | | | 9 | | 29 |
| Comfort | | 1 | | | 1 | 3 | | | 3 | 8 |
| Design | | 9 | 1 | | 9 | | | 3 | | 22 |
| Innovation | 3 | 9 | | | | | 9 | | | 21 |
| Safety | | 3 | 1 | | 1 | | | 3 | | 8 |
| Smart | | 1 | | | 3 | | | 3 | 1 | 8 |
| Driving Involvment | | 3 | 3 | | 3 | 3 | | | 3 | 15 |
| Personalization | 1 | | 3 | | | | | 3 | | 7 |
| Total | 5 | 27 | 12 | 10 | 28 | 6 | 9 | 21 | 10 | |

**Figure 5.** Independence matrix. Cells highlighted in green represent the winners for each specific category.

The highest values for the columns are also produced by performance and innovation, but driving is also involved, as can be seen by the fact that the highest values for the rows are produced by performance, design, and innovation.

The relationships between the various client product needs are clear once the dependency matrix has been established, but the relative significance of the different components is still unknown. In order to obtain this information, we must define the relative importance matrix, which has the same rows and columns as the previous one, but the question to answer is: "is the element in the row more important than the element in the column?" (Figure 6).

| | Price | Performance | Comfort | Design | Innovation | Safety | Smart | Driving Involvment | Personalization | Total | Importance |
|---|---|---|---|---|---|---|---|---|---|---|---|
| Price | 1 | 0 | 2 | 0 | 0 | 1 | 2 | 0 | 1 | 7 | 5.0 |
| Performance | 2 | 1 | 2 | 1 | 1 | 2 | 2 | 1 | 2 | 14 | 10.0 |
| Comfort | 0 | 0 | 1 | 0 | 0 | 0 | 1 | 0 | 1 | 3 | 2.1 |
| Design | 2 | 1 | 2 | 1 | 1 | 2 | 2 | 1 | 2 | 14 | 10.0 |
| Innovation | 2 | 1 | 2 | 1 | 1 | 2 | 2 | 1 | 2 | 14 | 10.0 |
| Safety | 1 | 0 | 2 | 0 | 0 | 1 | 2 | 1 | 2 | 9 | 6.4 |
| Smart | 0 | 0 | 1 | 0 | 0 | 0 | 1 | 0 | 1 | 3 | 2.1 |
| Driving Involvment | 2 | 1 | 2 | 1 | 1 | 1 | 2 | 1 | 2 | 13 | 9.3 |
| Personalization | 1 | 0 | 1 | 0 | 0 | 0 | 1 | 0 | 1 | 4 | 2.9 |

**Figure 6.** Relative importance matrix. Cells highlighted in green represent the winners for each specific category.

To answer the question, we use numeric values that can be the following:

- 0 if the element in the row is less important than the element of the column.
- 1 if they have the same importance.
- 2 if the element in the row is more important than the element in the column.

Finally, we must add the values in each row and normalize them to the maximum value we were able to determine. We can see that the most important elements with respect to the others are performance, innovation, design, and driving involvement.

### 2.2.2. Benchmarking Analysis and Top–Flop Analysis

These graphics demonstrate the current situation of the small sports car industry, which is, in our opinion, the market category for driving enthusiasts. They also demonstrate what these consumers demand. Because internal combustion engines dominate the top models in the market for small, entertaining sports vehicles, we believe there may be room for new engine types in this sector. The segment also offers a wide range of performances and, consequently, selling prices, ranging from thirty thousand to seventy-five thousand, which means that with some models it is possible to cross the threshold of one hundred thousand euros. This gives us the chance to introduce something novel at a price that we believe can still be competitive. We believe that the absence of a unique model on the market, together with the wide range of price options offered by this market, might be an advantageous consideration for the design of this new automobile. Following that, we conducted a study of the key rivals in the small sports car class while watching the market.

The analysis was conducted starting from the creation of a benchmark table containing the relevant data of each product in order to obtain a more complete technical overview. We then moved on to a top–flop analysis, in which the best and worst values, from our point of view, were selected for each technical category (Figure 7).

| | BMW Z4 sDrive30i | Audi TT 40 TFSI S-Tronic | Mazda MX-5 RF 2.0 SkyActiv-G | Porsche 718 Cayman S PDK | Mercedes-Benz SLC 300 | Alpine A110 Legende | Toyota GT86 2.0 | Nissan 370Z | Subaru BRZ | Lotus Elise Cup 250 | Innovation column |
|---|---|---|---|---|---|---|---|---|---|---|---|
| Lenght (mm) | 4324 | 4191 | 3915 | 4207 | 4133 | 4180 | 4240 | 4250 | 4240 | 3824 | 3824 |
| Width (mm) | 1864 | 1832 | 1735 | 1801 | 1810 | 1798 | 1775 | 1845 | 1775 | 1719 | 1864 |
| Height (mm) | 1304 | 1353 | 1236 | 1284 | 1301 | 1252 | 1320 | 1310 | 1320 | 1117 | 1117 |
| Wheelbase (mm) | 2470 | 2505 | 2310 | 2475 | 2430 | 2419 | 2570 | 2550 | 2570 | 2300 | 2300 |
| Power net (kW) | 190 | 145 | 135 | 257.5 | 180 | 185 | 147 | 241 | 147 | 183 | 257.5 |
| Torque net (Nm) | 400 | 320 | 205 | 420 | 370 | 320 | 205 | 363 | 205 | 250 | 420 |
| Curb weight (kg) | 1415 | 1360 | 1072 | 1385 | 1430 | 1100 | 1270 | 1496 | 1277 | 931 | 931 |
| Starting price | 53,100.00 € | 44,950.00 € | 34,500.00 € | 68,051.00 € | 49,950.00 € | 62,933.00 € | 31,700.00 € | 34,800.00 € | 32,990.00 € | 66,850.00 € | 31,700.00 € |
| Max speed (km/h) | 250 | 250 | 214 | 285 | 243 | 261 | 226 | 282 | 226 | 256 | 285 |
| 0-100 km/h (s) | 5.8 | 6.6 | 6.8 | 4.6 | 5.7 | 5 | 7.4 | 5.4 | 7.6 | 4.7 | 4.6 |
| Average cons (l/100 km) | 6.1 | 6 | 7 | 8 | 6.9 | 6 | 8.6 | 10.5 | 8.6 | 8 | 6 |
| CO2 emissions (g/km) | 139 | 137 | 154 | 167 | 158 | 138 | 181 | 248 | 181 | 175 | |
| | | | | | | | | | | | |
| Numero TOP | 1 | 2 | 0 | 4 | 0 | 1 | 1 | 0 | 0 | 4 | |
| Numero FLOP | 0 | 1 | 3 | 2 | 0 | 0 | 3 | 3 | 2 | 1 | |
| Delta | 1 | 1 | -3 | 2 | 0 | 1 | -2 | -3 | -2 | 3 | |

**Figure 7.** Benchmarking analysis; innovation column; top–flop analysis. Cells highlighted represent the winners (green) and losers (red) for each specific category.

In order to build the flagship product in the present market, the top values of reference, as of the most recent study, are provided in the innovation column. The number of top values and flop values for each vehicle were also listed in the table. The automobiles with the biggest delta values between top and flop are now the most creative.

Our goal is to obtain the best delta in order to overcome the Lotus Elise with a delta of 3.

### 2.2.3. What–How Matrix

Now, to understand what the best parameters are to improve, we are going to use another instrument, the what–how matrix [Tab 5]. This matrix enables us to compare the technical designers' responses to client demands. Therefore, we created another matrix that lists the needs in rows and the factors we may change to fulfill these requests in columns. Not all of the customers' demands, which were previously enumerated, will be included in this matrix; instead, only the requirements that received the highest values in the relative importance matrix will be included. Again, to quantify the relationship between the customer's request and the performance of the product, numerical evaluations are used:

- 0 means no relation;
- 1 means weak relation;
- 3 means medium relation;
- 9 means strong relation.

Once the weight of each box has been defined, the sums are calculated by rows and by columns. The maximum value of the row sum indicates which requirements are most affected by the parameters in the column, while the highest numbers among the column sums indicate which performance results would be more important in achieving the customers' requests (Figure 8).

| | LENGHT | WIDTH | HEIGHT | WHEELBASE | POWER NET | TORQUE NET | CURB WEIGHT | STARTING PRICE | MAX SPEED | 0-100(Km/h) | AVERAGE CONS (l/100km) | CO₂ EMISSIONS(g/Km) | |
|---|---|---|---|---|---|---|---|---|---|---|---|---|---|
| PERFORMANCE | 1 | 1 | 1 | 0 | 9 | 9 | 9 | 0 | 9 | 9 | 0 | 0 | 48 |
| INNOVATION | 0 | 0 | 0 | 0 | 3 | 3 | 9 | 9 | 3 | 3 | 9 | 9 | 48 |
| DESIGN | 9 | 9 | 9 | 9 | 0 | 0 | 0 | 0 | 0 | 0 | 0 | 0 | 36 |
| DRIVING INVOLVEMENT | 3 | 3 | 3 | 3 | 9 | 9 | 9 | 9 | 3 | 9 | 0 | 0 | 60 |
| | 13 | 13 | 13 | 12 | 21 | 21 | 27 | 18 | 15 | 21 | 9 | 9 | |

**Figure 8.** What–How matrix. The most important values that came out of this matrix are highlighted.

The customer requests most influenced by the studied parameters are:

- Performance;
- Innovation;
- Driving involvement.

The most influenced performances, on the other hand, are:

- Power net;
- Torque net;
- Curb weight;
- 0–100(Km/h).

In the top–flop analysis, we noticed that we at least four parameters required innovation, and from the what–how matrix above, we obtained these four; now, in the product architecture, we will see how we work on these parameters, while also trying to introduce even more innovation.

## 3. Results

### 3.1. Product Architecture

The range architecture, which is the technological advance project, and the schematization of the layout of the technical function components, which define the range of the new product, must be defined before the SDE technique can be used. As we are dealing with a compact sportscar, we will examine the market segment in a similar manner to the way in which we conducted the benchmarking analysis in order to determine the overall dimensions [33].

We notice that the maximum length is that of the Porsche 718 Cayman S, which measures 4397 mm, while the minimum length is that of the Lotus Elise Cup 250, which measures 3824 mm. The same can be seen for the width where the maximum width is

that of the BMW Z4, which is 1864 mm, while the minimum is that of the Lotus Elise Cup 250, which measures 1719 mm. Regarding the height, the Lotus Elise has the lowest measurement, namely 1117 mm, while the highest measurement is that of the Audi TT, which is 1355 mm.

We determined the total size of our automobile by analyzing the dimensions of the seats, engine, and hydrogen tank. The overall measurements were 4100 mm long, 1200 mm high, and 1850 mm wide.

### 3.2. Innovation

### 3.2.1. Materials

The weight is another crucial factor; the Lotus Elise 250 Cup has the lowest weight in its category, and we intend to maintain that lightness by utilizing aluminum and carbon fiber for the chassis and recycled polypropylene for the body. Sandwich composites are structural materials in layers composed of two layers of composite polymer at the ends and a replenishing polymer inside. This composition allows us to produce large parts with good mechanical properties and low weight. A problem related to using carbon sheets is the anisotropy due to the intrinsic nature of the material; usually, to overcome this problem, the various layers are spread at different angles, typically 0-45-90-45-0 degrees.

The production process of a carbon chassis is divided into five phases:

1. 2D shaping of sheets;
2. 3D shaping in mold;
3. Resin injection in the press putting together all the pieces (CFRP and Al. parts);
4. Removing the monocoque chassis from the press;
5. CNC machining to refine and open holes.

### 3.2.2. Power and Torque

Regarding the engine for this study, we chose as an example a 90° V6 engine, such as the one shown in (Figure 9).

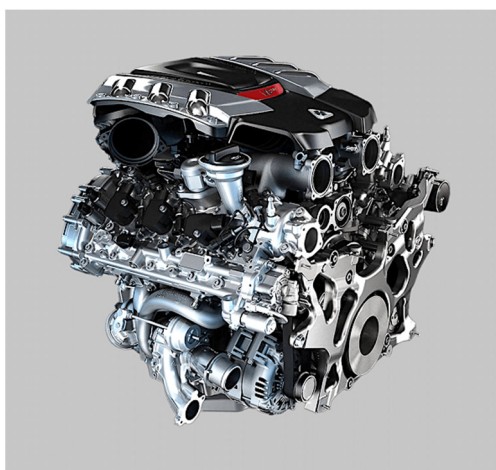

**Figure 9.** 90° V6 Engine.

As we are using hydrogen, we do not need to apply much demand on the specific power. In order to prevent the exceptionally high pressures that may encourage knock and explosion, we would also need to lower the compression ratio. The example engine's output in this arrangement would be between 350 and 400 horsepower.

### 3.2.3. Considerations and Criticisms on the Choice of Hydrogen and HYICE

One of the most potent resources we have on the earth is hydrogen. It has a lot of energy inside that can be used, but hydrogen also offers a lot of challenges. Due to this decision, we must deal with storage issues, lubrication issues, and safety issues. In this

section of the project, we will discuss in detail the safety issue that most concerns us, the reason for hydrogen's propensity to explode under certain conditions [34].

The temperature range between 675 K and 850 K is where pressure determines whether hydrogen will explode; it cannot burst below 675 K and can explode above 850 K [35]. To understand how this problem was born and how to solve it, we need to rapidly analyze hydrogen's combustion mechanism. Inside this mechanism, we have a lot of reactions, but the crucial reactions are those of the chain starting and the chain branching.

Hydrogen's combustion mechanism has essentially two initiating chain reactions:

$$M + H_2 \rightarrow M + 2H^{\bullet}$$

$$H_2 + O_2 \rightarrow HO_2{}^{\bullet} + H^{\bullet}$$

The second reaction produces a bigger radical, which provides slow reactions; so, we consider the first one. Then, we have all the propagation reactions and, at the end, the chain branching.

One of the propagation reactions is:

$$H^{\circ} + O_2 \Rightarrow OH^{\circ} + O^{\circ}$$

When we have the presence of a particular molecule, which we call M, this reaction takes a different path:

$$H^{\circ} + O_2 + M \Rightarrow HO_2{}^{\circ} + M$$

Molecule M only has the task of absorbing the bond's energy; so, it remains unchanged in the reaction.

As we can see, we returned to the radical that we described in reaction 2; this radical, as we have already said, is very big and provides a slow reaction; so, we can consider this as a chain branching reaction. This reaction prevails on the one before when the pressure rises; so, at a certain pressure valor, we have no more reactivity, but when the pressure rises even more, the concentration of the radical $HO_2{}^{\circ}$ becomes so high that this slow reaction becomes no longer negligible. This means that this reaction is no longer a chain branching, but it becomes a chain propagation reaction that can lead to explosion. Now imagine the rising temperature without changing pressure; the first reaction follows the Arrhenius equation, while the second one does not. This means that a rise in temperature corresponds to an increase in the speed of the first reaction to the detriment of the second one until we arrive again in the explosive condition.

Here, we have a diagram that describes hydrogen's combustion behavior and sums up all the concepts written before.

After this analysis, we understand that we need to work under 850 K and to control the pressure between 675 K and 850 K (Figure 10).

### 3.2.4. How to Obtain Hydrogen

Although there are several ways to obtain hydrogen, not all of them have no environmental impact. Currently, we know four ways to acquire hydrogen:

1.  Methane reforming;
2.  Coal gasification;
3.  Algae and bacteria;
4.  Electrolysis.

Without going into too many details the first two methods produce carbon dioxide as waste. Considering that the biggest global goal now is the reduction in $CO_2$, these are not viable methods.

The third method does not produce polluting waste, but it is very sensitive to pH and to environmental conditions; so, it is difficult to produce a large quantity of hydrogen with this process.

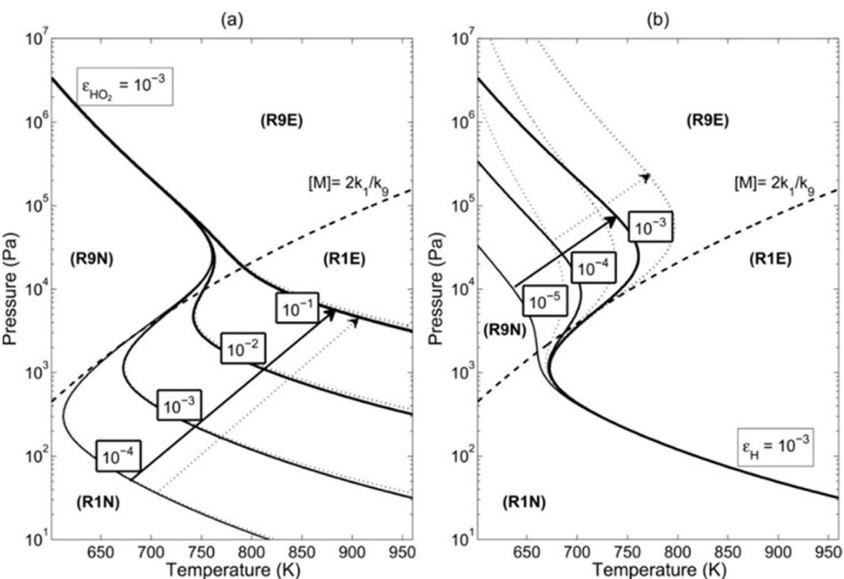

**Figure 10.** Explosion limit for a stoichiometric hydrogen–oxygen mixture (**a**,**b**).

The fourth one is the most desirable; it can produce hydrogen using electric energy. It is an expensive method and of course its impact on pollution depends on the source of electricity. However, since we are searching for a method to store the energy produced by renewable sources, we can consider using electrolysis as a process to store energy and hydrogen as a tank in which to store it.

In conclusion, hydrogen is a difficult matter, but with huge benefits to exploit: we could solve the pollution problem through hydrogen. Furthermore, it is important to consider European incentives for HYICE development.

### 3.3. Two-Dimensional Architecture and Sketching

Having established the overall dimensions of our car (Figure 11), we can proceed with the sketching in accordance with the SDE method. The sketching phase was conducted following four main design styles (Figure 12), which are the retro, the natural, the stone, and the advanced (Donnici et al., 2020 [32]).

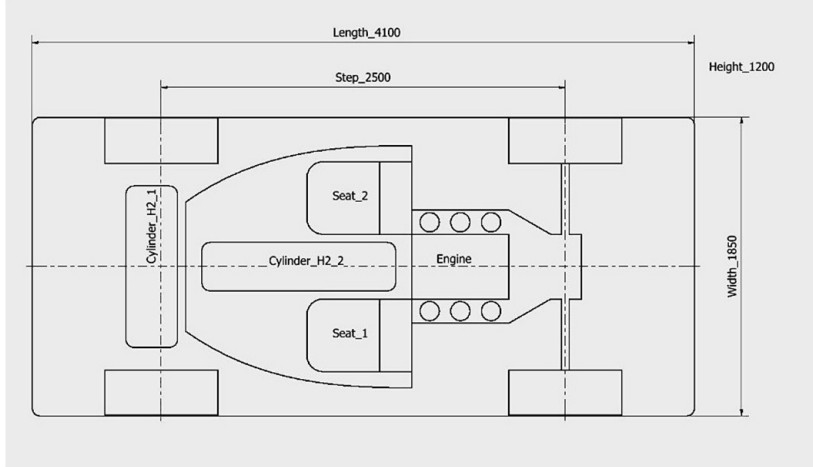

**Figure 11.** Overall dimensions.

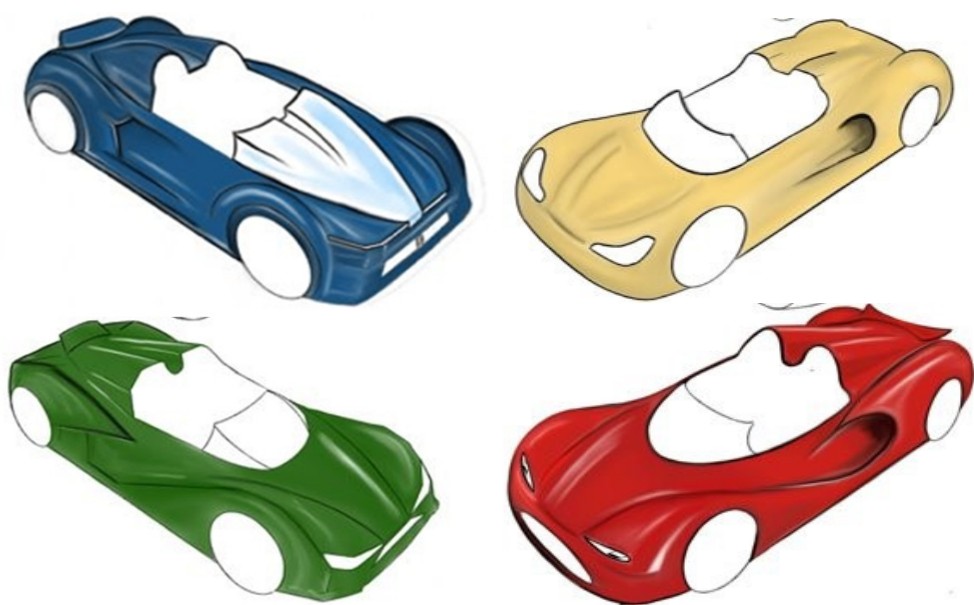

**Figure 12.** Designs.

### 3.3.1. Retro

The retro style is inspired by the old cars, where we empathize with the characteristics of the cars made in past times, from the 1950s' until the end of 1980s'.

### 3.3.2. Stone

The stone style has sharp lines and massive forms; it is mainly used for SUV cars.

### 3.3.3. Natural

The natural style is characterized by smooth curves, inspired by proportions in nature.

### 3.3.4. Advanced

The advanced style is characterized by futuristic curves, the presence of reflections, sharp lines, aggressive appearance, and a flashy style.

### 3.3.5. Blueprints

From sketches, we created blueprints which gave us the possibility to model the surface of the car.

Following the production of the blueprints for each of the four designs (Figure 13), we went on to create simplified 3D models of each style in order to conduct a preliminary CFD study and to determine which concept was the best from an aerodynamic standpoint. The drag coefficient that we calculated using a CFD analysis on the software "Ansys" served as our benchmark.

The results are the following:

- Retro: 0.49;
- Stone: 0.39;
- Advanced: 0.50;
- Natural: 0.45.

As can be seen from the CFD study, the stone and natural proposal has the best drag coefficients; hence, for the final model of our automobile, we decided to combine both styles (Figure 14).

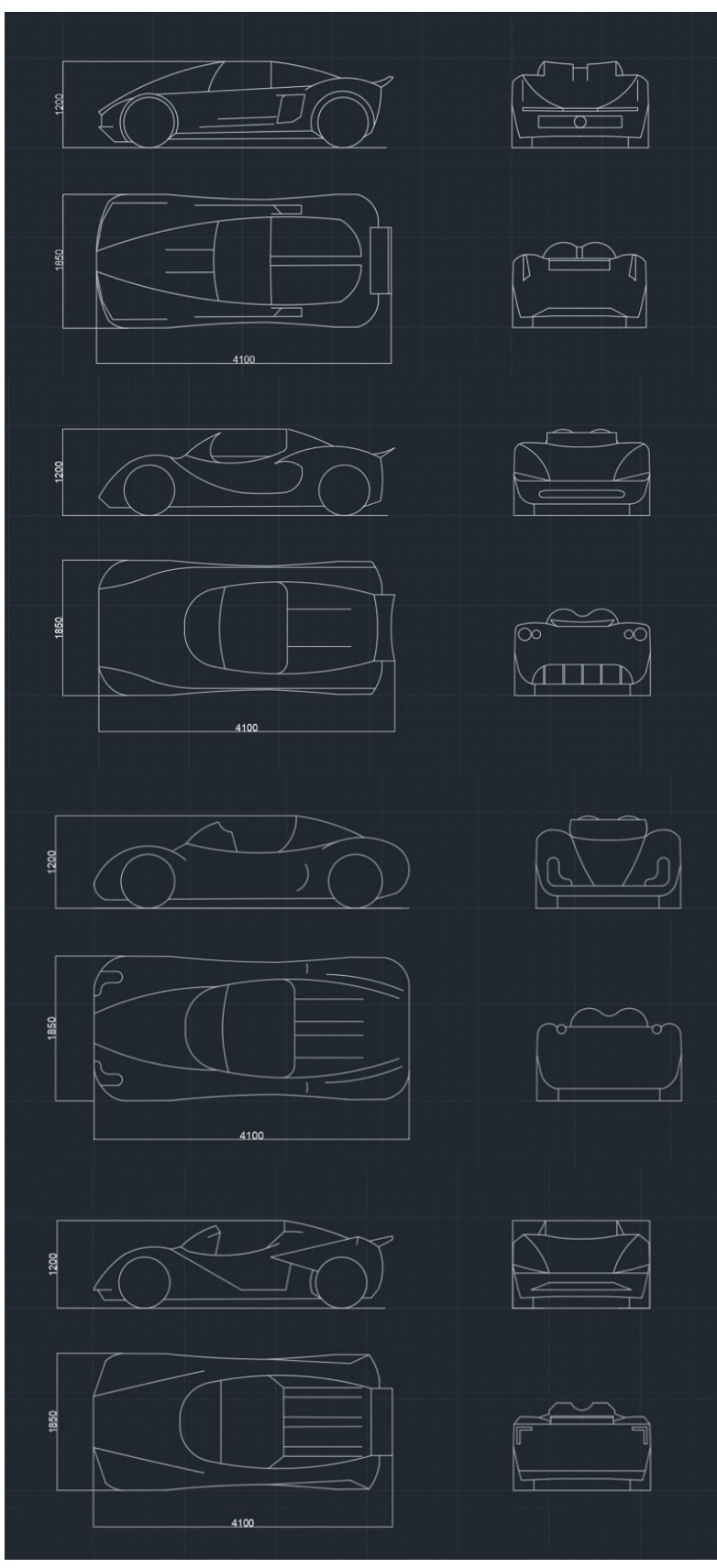

**Figure 13.** Blueprints.

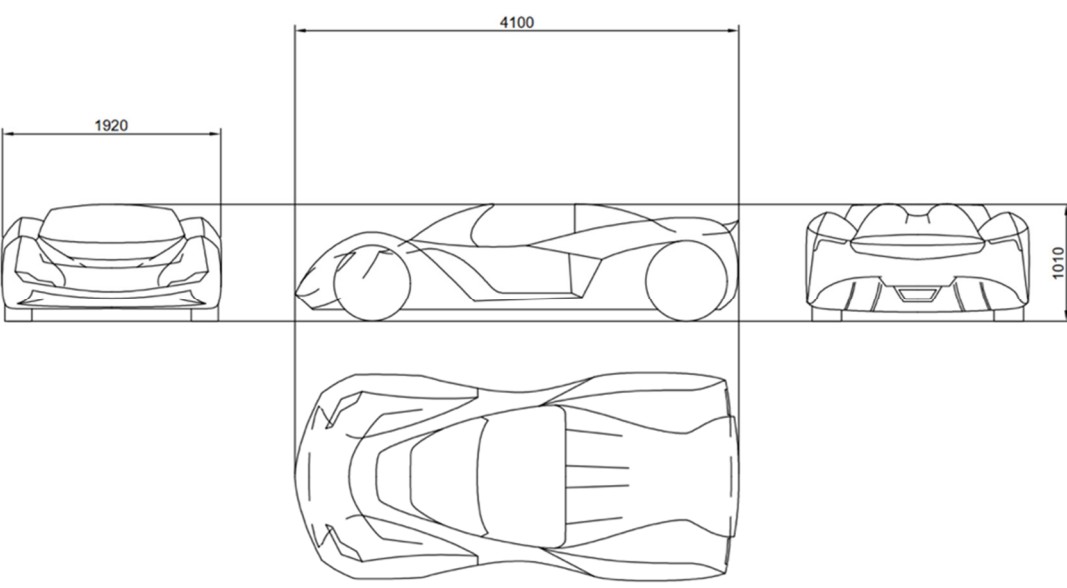

**Figure 14.** Blueprints final model.

*3.4. Three-Dimensional Architecture*

Once the 2D architecture was established, we created a chassis with an aluminum front and rear subframe and a carbon fiber shell [36–38]. A mid-engine mechanical configuration anchored to the rear subframe was chosen for optimal drivability, and the placement of hydrogen tanks within the overall structure was also considered (Figure 15).

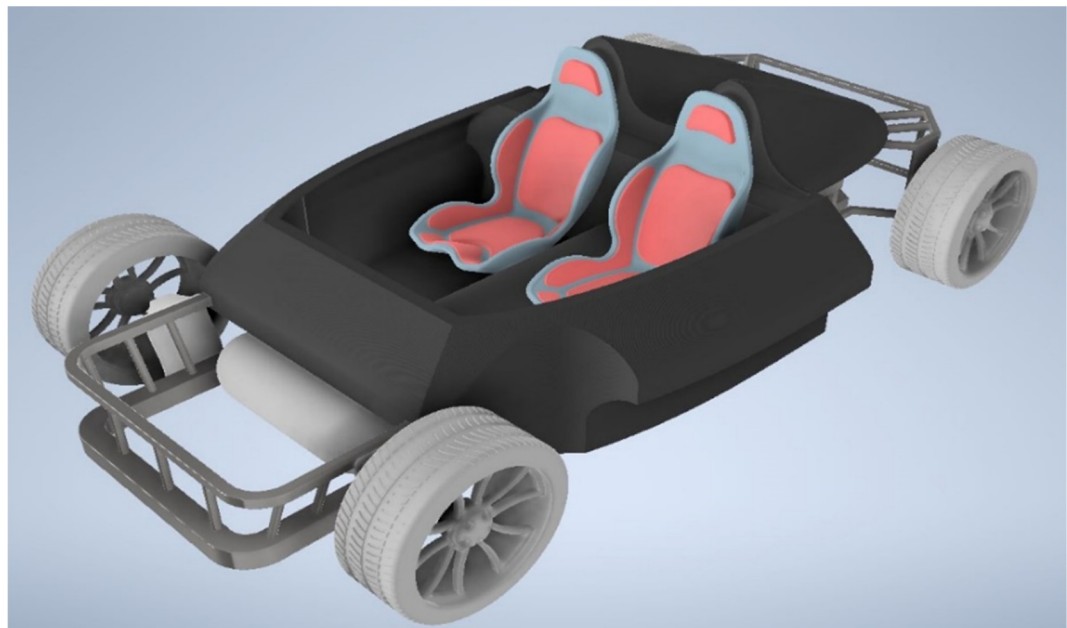

**Figure 15.** Chassis.

*3.5. Three-Dimensional Surface*

After we performed the CFD analysis, we started to model our final 3D surfaces (Figure 16) using Alias. The main constraint aspect of this part is that we had to design the 3D surface of the car around the chassis that had already been modeled.

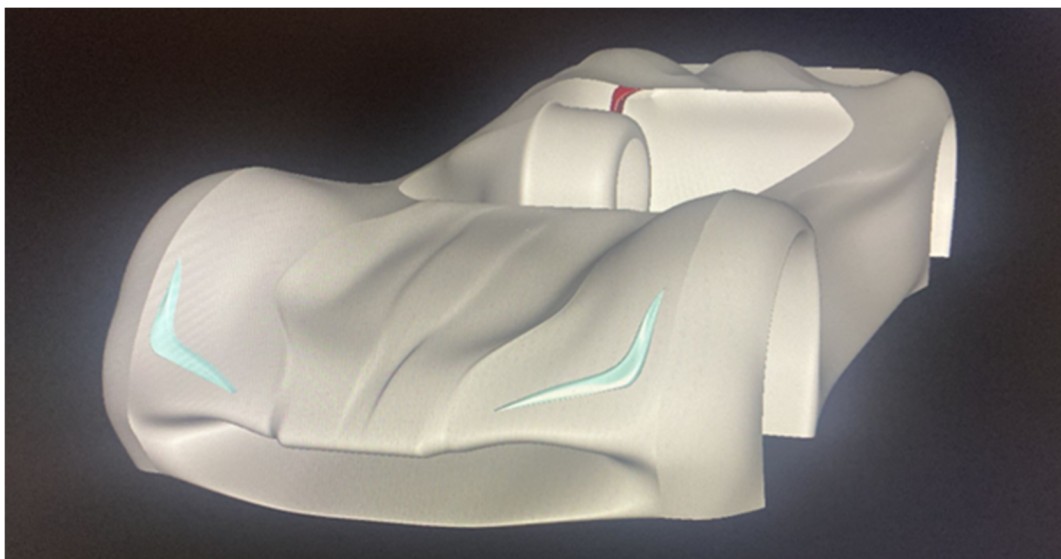

**Figure 16.** Surface.

*3.6. CFD Analysis of the Final Model*

After designing the car's 3D surface, we were able to run a new CFD simulation to evaluate the vehicle's actual drag coefficient [39,40] (Damjanović et al., 2011). We can see the air speed streamline in Figure 15. The final drag coefficient we obtained was 0.28 (Figure 17) which was comparable with that of the cars of the same segment already on the market. The result was considered satisfactory, and the analysis provided insight into the behavior of flows around the car body. This allowed strategic changes to be made to optimize the shape details of the car.

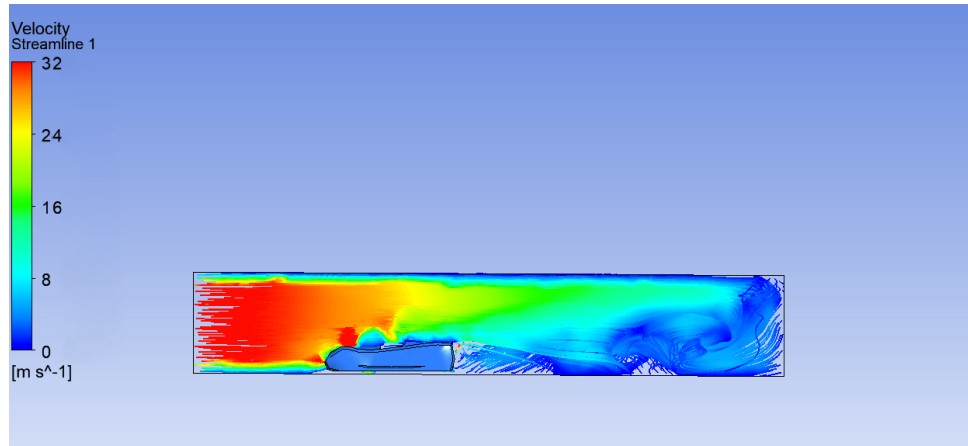

**Figure 17.** Air speed steam line.

*3.7. Rendering*

The creation of the model using 3D CAD was followed by the creation of the photorealistic renderings and photoinsertions in order to better understand the shapes and proportions of the final product in a virtual environment (Figures 18 and 19). The errors found at this stage led to the modification of the 3D model, allowing arrival at the next stage, that of the physical prototyping with fewer errors and imperfections, thus decreasing the time and costs.

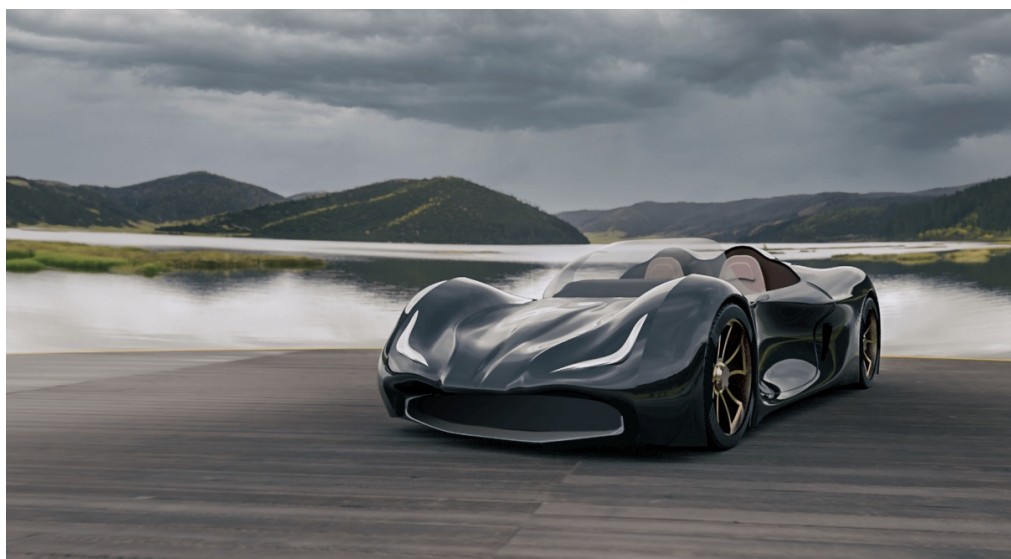

**Figure 18.** 3/4 front view.

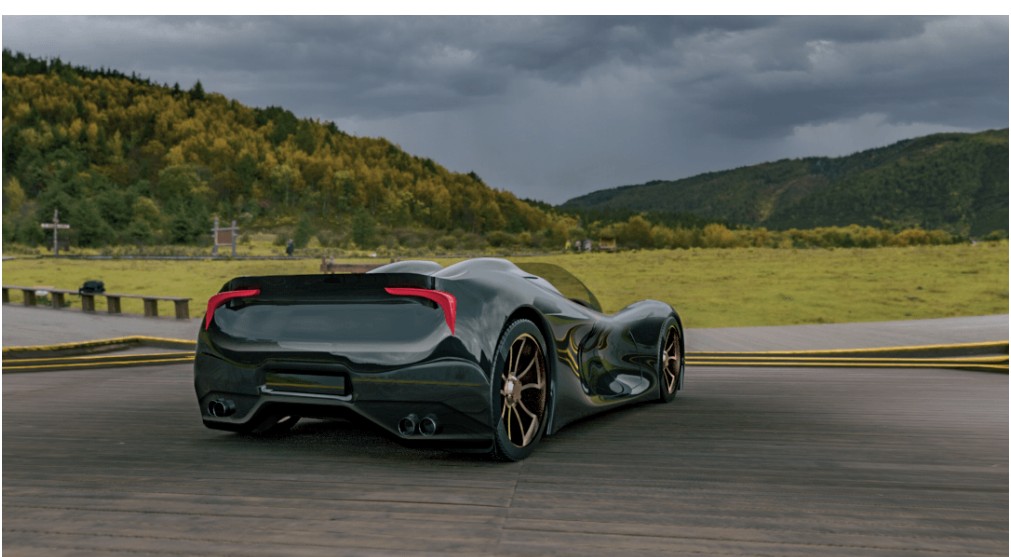

**Figure 19.** 3/4 back view.

### 3.8. Prototyping

For the prototyping model, we used the FMD 3D printing technology, post-processed with grouting and painting (Figure 20) [41].

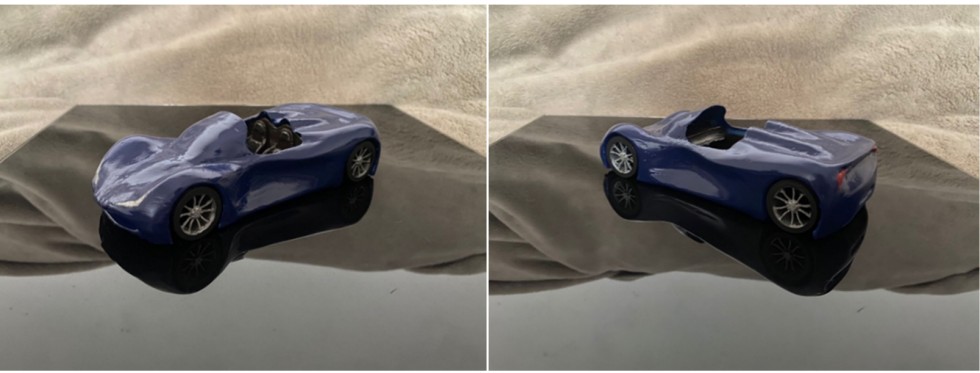

**Figure 20.** Prototype.

## 4. Discussion

In order to meet the new regulations imposed by many governments to reduce pollution and avoid greenhouse gases due to vehicles, this project was created with the idea of researching the possibility of hydrogen internal combustion engine-driven vehicles. This would improve the carbon emission impact of future cars without using an electric-driven vehicle.

The goal was to guarantee an alternative to electric vehicles for all of those who wanted to maintain internal combustion engine driven vehicles. There were several challenges that had to be overcome, such as the hydrogen storability, due to its low density; the revisitation of an already existing engine in order to convert it to hydrogen; and the use of lightweight materials such as carbon fiber for the chassis monocoque.

As is possible to see in the image above, we combined two styles to create our final stylistic proposal. CFD and FEM analyses were then utilized to confirm the engineering design. We used the IDeS approach to develop the vehicle's styling. In the end, we were able to achieve some intriguing engineering numbers, such as a drag coefficient of 0.28, which is on a par with the top sports vehicles now available.

Even though we are aware of the advancements that technology has yet to make, we are certain that a project built on these foundations may be distinctly creative and admired with an eye toward the future of sports mobility. The production and transportation of hydrogen is a topic that is more relevant than ever, and we believe that a compact sports vehicle, which places a premium on driving enjoyment, is the ideal fit for hydrogen due to its unique qualities. As sports car lovers and fans of sports vehicles in general, we are certain that if a solution such as the one we offered were to become reality on a broad scale, it would be a positive step forward for the industry as a whole.

## 5. Conclusions

In terms of the next advances, we considered a few crucial elements needed to move toward the creation of a functioning prototype. In terms of stylistic development, we believe we are well advanced; consequently, it will be essential to improve:

- power development;
- re-design and adjustment of the chassis based on future developments of the propulsion and hydrogen storage system;
- development of vehicle dynamics;
- production strategies;
- any changes based on market trends.

In the future, the production and distribution of hydrogen will undoubtedly pose the greatest challenge. These issues can, however, be optimistically overcome with the introduction of specific state regulations and the allocation by governments of funds and resources for the development of technology and distribution. An experimental method of transporting hydrogen is to create a mixture of hydrogen and natural gas within the existing natural gas distribution network. The technical specification UNI/TS 11537: 2019 sets a technical acceptability limit equal to 1% of the volume of hydrogen in the biomethane that can be fed into the network; in fact, experiments have been conducted that have shown the possibility of successfully creating a 10% mixture of hydrogen in natural gas. In February 2020, the Italian Independent Regulatory Authority for Energy, Networks and the Environment ("ARERA") launched a public consultation process in relation to pilot projects to optimize the management and innovative uses of the natural gas transportation and distribution networks already in existence.

According to ARERA, P2G plants will become economically sustainable starting from 2030, mainly because this technology requires large amounts of low-cost electricity to be competitive. In this document, ARERA puts the spotlight on the possibility of developing pilot projects relating to the integration of renewable gases in networks and applications of P2G and power-to-hydrogen—P2H technologies. The latter are essential so that the hydrogen produced through electrolysis can also be used as a storage vector

to produce electricity again with reversible fuel cell systems (power-to-power—P2P). The most accredited vision with regard to the transport sector is that hydrogen will initially find space, for example, in the field of public transport, especially for long-distance routes, in commercial freight transport fleets and in parts of the railway network that are not electrified. The hydrogen refueling stations, at an early stage, may be managed by regional or local transport companies, based on a clear analysis of the fleet demand and on the different requirements for light and heavy vehicles.

*F.E.M. Analysis*

A more thorough FEM study of the chassis than the one we carried out (Figures 21 and 22) will be required in order to validate and optimize the model. The key research that has to be conducted is on stiffness, in order to enhance the car's dynamic behavior. We intended to carry out a bending study at this stage of the project that merely mimics the static forces operating on the chassis [42,43]. The study was carried out by applying a total load of 15,000 N to the center of mass of the chassis, which was equal to the static force operating on the car, while accounting for the engine's position at the back. Figures 21 and 22 below illustrate some possible FEA results.

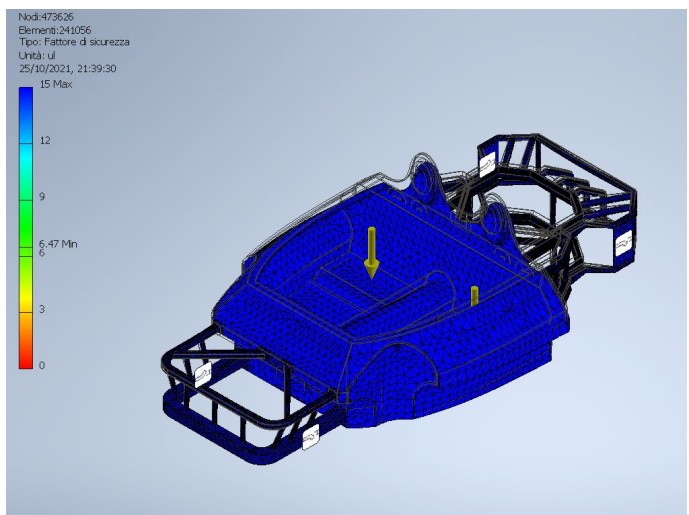

**Figure 21.** Safety factor. Arrows indicate the points where the force is applied.

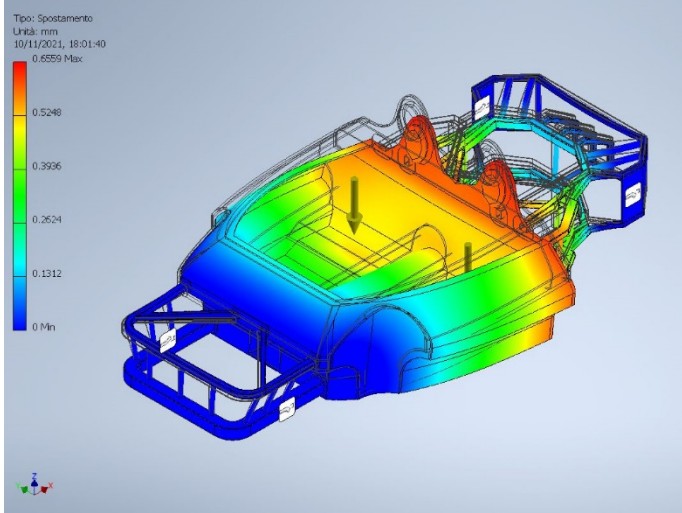

**Figure 22.** Total deformation. Arrows indicate the points where the force is applied.

**Author Contributions:** G.G. was in charge of the general project reviews and management of the paper's implementation; M.C., P.M., M.R. and I.V. handled the development of the project from the initial stages and throughout its duration; L.F. reviewed the group work along all stages of development and gave overall approval. All authors have read and agreed to the published version of the manuscript.

**Funding:** This research received no external funding.

**Acknowledgments:** The materials and machines used for the developing of the prototypal phase were granted by the DIN—Department of Industrial Engineering at Alma Mater Studiorum Università di Bologna.

**Conflicts of Interest:** The authors declare no conflict of interest.

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
