# Peer review of "Use of IDeS Method to Design an Innovative HYICE Sportscar"

_inventions, doi:10.3390/inventions8030075_

Round 1

Reviewer 1 Report (Previous Reviewer 2)

The article has already been reported in a similar form.

The research method should include an experimental study, mainly focusing on assumptions in the form of a case study.

It cannot be called innovative. Nalkezy to make a comparison with existing solutions, also in other European countries.

The article mentions the developing electric motors and hydrogen-powered electric motors. No mention of CO2 emissions? I mean the entire technological process related to the supply of significantly larger amounts of traditional raw materials to the network.

What is the energy efficiency of the proposed solution?

Author Response

Dear Reviewer, Thank you very much for your valuable feedback.
We have improved some points in the paper to better clarify some aspects you mentioned. We understood your comments about the motorization, however, the main focus of the study is not on the hydrogen engine, but on the IDeS methodology that led to the development of the concept. The in-depth study on the motorization and all aspects related to it was not purposely done because it was felt that it could be misleading to the actual topics of this research. In fact, the focus of the study is on design methods and processes that, taking into account different technical and technological opportunities (such as hydrogen motorization) manage to systematize the data in an efficient method to arrive in short order at the definition of a convincing design concept, on which only then to go on to carry out a more technical in-depth study such as the one you suggested.

Reviewer 2 Report (New Reviewer)

The research content of this paper is very interesting. 

Comments:

(1) Different studies on this topic are available before. What is the difference from the others? It must be emphasized.
(2) In Section 1. Introduction, relevant research progress needs to be summarized more comprehensively. In addition, contributions need to be listed at the end of the introduction which could be friendly to the readers.
(3) In the case study, it is not sufficient to explain the results. They need to be disclosed.
(4) The numbers in Figs. 17-18 are not clear enough. Furthermore, It seems that some inconsistencies in the results are available.

Author Response

Dear reviewer, Thank you very much for your suggestions.

Your recommendations have been included into our paper, with the most significant modifications highlighted in yellow.

(1)-(2) To clarify several issues you mentioned, we've changed or added to some passages.

(3) We added explanations at several critical spots and rewrote other lines to help clarify certain concepts.

(4) To obtain more intriguing results, we conducted a more extensive investigation. We then changed that portion of the report, modifying the figures as well.

Thank you

Reviewer 3 Report (New Reviewer)

The authors tried to present all aspects of a novel hydrogen sport car. From high level energy consideration to design of the engine and the chassi FEA. As it would be expected almost all parts are incomplete. Despite of this the message of considering this technology for some part of transportation is clear. For this reason I would propose to improve the two engineering parts, namely engine technology, safety requirements and FEA and resubmit the paper for further review.

Author Response

Dear Reviewer, Thank you very much for your valuable feedback.

To better address some of your issues, we have updated several parts in the document. We are aware of your engineering-related concerns, however the study's primary focus is on the IDeS approach, which helped to build the concept, not on the hydrogen engine. Since it was believed that doing so may be confusing to the real issues of this research, an in-depth investigation of motorization and all related features was not purposefully conducted. In fact, the study's main focus is on design methodologies and processes that, while taking into account various technical and technological opportunities (such as hydrogen motorization), are able to efficiently systematize the data and quickly arrive at the definition of a compelling design concept, from which only further technical in-depth research like the one you suggested would be possible.

Round 2

Reviewer 1 Report (Previous Reviewer 2)

The article has been significantly improved. There is a lack of literature review of existing research studies. The described research method should be supported by an empirical model. Please explain to what extent the research overlaps with existing solutions and why attention has been focused on these car models? Simulations should focus on common brands.

Author Response

Thank you for your valuable feedback. We added some new references related to literature review about hydrogen engines and applications. The research method described is supported by the IDeS methodology, which has it's own description and references. As described, this method has some overlaps in the engineering phase, but also integrates a lot of design and styling processes within it, making it more complete and useful for complex applications like automobile design, like the one described in this article. If you think that those aspects are not clear enough inside the document, we can improve them further.

Best Regards

Reviewer 3 Report (New Reviewer)

The authors revised the manuscript in several places, responding to the comments. No further questions exist.

Author Response

Thank you for your review.

Round 3

Reviewer 1 Report (Previous Reviewer 2)

The article in the review must reflect the actual state of research already performed by scientists in other centers. The research methodology is based on a review of non-real data. Data should be presented and discussed. Conclusions should reflect the tests performed on the models.

Author Response

Thank you very much for your valuable feedback.

The article has been edited, with the "Introduction" section almost completely rewritten, new information added to better clarify the points highlighted by your review, and several other paragraphs within the doument modified. Approximately 20 new references were also included to give a better framing of the work. The literarature review has been updated as well.

Best Regards,

Giulio Galiè

This manuscript is a resubmission of an earlier submission. The following is a list of the peer review reports and author responses from that submission.

Round 1

Reviewer 1 Report

This is the review of the manuscript entitled „Use of IDeS Method to Design an Innovative HYICE sportscar”. The authors present an interesting topic, being in line with the mission of the Journal.

There are anyway some points that are to be addressed and taken care of:

1) The abstract section is missing some qualitative/quantitative main research findings.  Avoid general discussions and concentrate on the aim and objectives of this paper.

2) The novelty of the work must be clearly addressed and discussed, compare your research with existing research findings and highlight novelty.

3) The main objective of the work must be written on the more clear and more concise way at the end of the introduction section.

4) Introduction section must be written on more quality way, i.e. more up-to-date references addressed. Research gap should be delivered on more clear way with directed necessity for the conducted research work.

5) The manuscript appears to be a collection of data. There is no methodology or research method that leads to certain results. It is not clear to the reader what are the input data for formulating the problem addressed to research by the authors and what are the results obtained after applying a research methodology/method. The information in the manuscript is difficult for the reader to follow, with many generalities presented and the lack of critical analysis, but also with very few advances in the field. ... it is difficult for the reader to understand what the authors want to claim.

6) The conclusions should highlight the main dedicated results based on the research work that the authors wish to disseminate with a direct connection with the main manuscript. Based on the critical analysis and discussions offered in the study, the authors should emphasize 3-5 main points as important claims resulting from their present work.

7) Outline future research directions quantifying main research findings.

8) Some presentation and language issues:

- L149 „2.1.1. Subsubsection” ... establish an appropriate title for this subsection.

- Define all notations that are used where the concept first appears in the text, not repetitively.

- The quality of the figures must be sharper.

- There are some typos in the manuscript. Please double-check.

- references must respect the style required by MDPI.

Author Response

Dear Reviewer, thanks for your suggestions.

We changed the abstract to better match the purpuse of the article and to highlight the methods used and the innovations we provide. We inserted Materials and Methods subsection to make the methodologies used better understood and to align with the MDPI standard. We have included references to data sources and tables. We have reworked the conclusions based on your suggestions to make them clearer. We have tried to argue more about the methodologies used and why they are applied in the given contexts, so that the paper does not appear like a collection of data, but to make the why and how of their use and application understood. The references have been aligned with the MDPI standard.

We double-checked minor English and writing errors and improved the quality of all pictures.

Reviewer 2 Report

Suggestions for Authors

Apart from the research results, the abstract should take into account the purpose and the scientific method.

The publication requires editing in line with the requirements of the Energies journal.

For example, references to citations, citing figures, diagrams in the text, description of tables.

There is no subsection describing Materials and Methods. The described concept is insufficient.

Figure 4 and Table 1 no data, sources of their origin. Please discuss and clarify.

The lists of cars in Table 4 are not clear. Where does this data come from?

In the proposed model, the technical specification regarding performance, technical parameters and real implementation prospects should be indicated.

Please specify the references according to the MDPI requirements.

Author Response

Dear Reviewer, thanks for your suggestions.

We changed the abstract to better match the purpuse of the article and to highlight the methods used. We inserted Materials and Methods subsection to make the methodologies used better understood and to align with the MDPI standard. We have included references to data sources and tables. The car data in Figure 4 are derived from the respective official websites for each car. The references have been aligned with the MDPI standard.

We double-checked minor English and writing errors

Round 2

Reviewer 2 Report

Corrections were made to a slight degree. Bark of detailing the research carried out in accordance with the comments of the reviewer. These substitutions should be clearly marked